# ZnO-Loaded Graphene for NO_2_ Gas Sensing

**DOI:** 10.3390/s23136055

**Published:** 2023-06-30

**Authors:** Mohamed Ayoub Alouani, Juan Casanova-Cháfer, Frank Güell, Elisa Peña-Martín, Sara Ruiz-Martínez-Alcocer, Santiago de Bernardi-Martín, Alejandra García-Gómez, Xavier Vilanova, Eduard Llobet

**Affiliations:** 1Microsystems Nanotechnologies for Chemical Analysis (MINOS), Universitat Rovira i Virgili, Avda. Països Catalans, 26, 43007 Tarragona, Spain; 2ENFOCAT-IN2UB, Universitat de Barcelona, C/Martí i Franquès 1, 08028 Barcelona, Spain; 3Gnanomat S.L. C/Faraday, 7. Parque Científico de Madrid, 28049 Madrid, Spain

**Keywords:** graphene, zinc oxide, gas sensor, NO_2_ detection

## Abstract

This paper investigates the effect of decorating graphene with zinc oxide (ZnO) nanoparticles (NPs) for the detection of NO_2_. In this regard, two graphene sensors with different ZnO loadings of 5 wt.% and 20 wt.% were prepared, and their responses towards NO_2_ at room temperature and different conditions were compared. The experimental results demonstrate that the graphene loaded with 5 wt.% ZnO NPs (G95/5) shows better performance at detecting low concentrations of the target gas than the one loaded with 20 wt.% ZnO NPs (G80/20). Moreover, measurements under dry and humid conditions of the G95/5 sensor revealed that the material is very sensitive to ambient moisture, showing an almost eight-fold increase in NO_2_ sensitivity when the background changes from dry to 70% relative humidity. Regarding sensor selectivity, it presents a significant selectivity towards NO_2_ compared to other gas compounds.

## 1. Introduction

Air quality, which is strongly correlated with public health and environmental problems, is a major societal concern. According to the World Health Organization (WHO), annually, 7 million premature deaths occur worldwide [1] caused by air pollution. In this sense, the European Environment Agency (EEA) recorded 307,000 deaths in Europe in 2021 [2]. The main atmospheric pollutants, as stated by WHO, are NO_2_, CO, SO_2_, and O_3_ with limits of exposure per day (updated in 2021) of 13 ppb, 4 ppm, 15 ppb, and 51 ppb, respectively [3]. Due to the rapid increase of the fossil fuel industry, and the number of powerplants and advancements in the automobile industry in recent years, NO_2_ has become the most common hazardous pollutant [4]. Moreover, the presence of this red-brownish and highly reactive gas in the air, even with low concentrations, can result in various dangerous effects on humans, such as severe throat infections and asthma [5]. Meanwhile, long-time exposure to NO_2_ is known to be very serious as it can cause permanent organic lesions in the lungs or even death [6]. Furthermore, this toxic gas is also dangerous to the global ecosystems and the environment, since it has adverse negative effects on water, soil and atmosphere [6]. To limit the danger of these airborne pollutants, in-field monitoring is mandatory. This is where gas sensors come in handy as a tool to achieve this goal.

Many types of gas sensors emerged intending to continuously monitor trace level concentrations of the previously mentioned gases, such as gas chromatographic systems [4], as well as electrochemical [5], conductometric, potentiometric [6], surface acoustic wave (SAW) sensors [7,8,9] and chemoresistive devices [10]. Among these different types, chemoresistive devices have emerged as an interesting option owing to their straightforward fabrication, ease of operation, miniaturization, and low production cost [11]. Chemoresistors consist of a conductive or semi-conductive chemically sensitive film deposited between two metal electrodes that shows a change of resistance when exposed to a target chemical analyte [12]. The most widely known and used nanomaterials in recent years are metal oxides (MOX) [13]. They present some advantages, such as small size, ease to manufacture with low costs, simple measuring electronics, and short response time to different gases such as NO_2_ [14,15]. However, despite being useful and effective for a long period, several drawbacks such as poor selectivity, baseline drift after a long period of usage, high sensitivity to humidity, and high-power consumption are still an issue [16]. In this regard, it is relatively easy to find MOXs that need to be heated to high temperatures up to 400 °C [17]. For detecting NO_2_ at room temperature, other alternatives to MOX sensors have been investigated, such as chemiresistors. Among these alternatives, it is worth mentioning graphene [18], MXenes [19], transition metal dichalcogenides [20,21,22] or a combination of these [23]. Graphene has attracted great research efforts as an alternative to overcome some of the limitations experienced with MOX sensors. Indeed, graphene shows some interesting properties to be implemented in chemoresistive gas sensors, such as a large specific surface area and high-carrier density and mobility in the near ambient temperature [24]. But not limited to this, different approaches were reported to further increase its sensing properties. One of them is the graphene functionalization with materials known for their sensitivity towards common toxic gases and especially NO_2_, in order to boost the graphene-based sensor performance. In particular, it was proven that the addition of MOX to graphene has resulted in substantial enhancement of its properties and sensing abilities, with ZnO as the leading MOX for this type of addition. Furthermore, ZnO is a semiconductor with a band gap of 3.37 eV and exciton binding energy of 60 meV, which makes it very useful for making stable devices. It is also considered thermally and chemically stable, which makes it also perfect for the production of devices for a long period of use [25]. But most importantly, the fact that ZnO has a good response towards different toxic gases, as well as having a low cost, has made it widely studied in gas sensing applications [26]. Recent research reporting the mixing of ZnO and graphene can be cited, e.g., Park Jing et al. who fabricated exfoliated graphene decorated with ZnO quantum dots sensors for enhanced methanol sensing at room temperature [27]. Additionally, Brigida et al. have studied the possibility of the improvement of NO_2_ detection by making graphene decorated by ZnO nanoparticle gas sensors [28].

In this work, we have synthesized graphene/ZnO nanocomposites using a novel procedure that uses moderate temperatures and pressures jointly with non-toxic solvents, allowing the mass production of the nanocomposites. These nanocomposites have been employed to manufacture chemoresistive gas sensors, which have been characterized. Their performance is then compared with the one of previously reported sensors making use of similar materials.

## 2. Materials Preparation and Methods

### 2.1. Nanocomposite Synthesis and Deposition

G80/20 and G95/5 nanomaterials, with a weight ratio of 80:20 and 95:5, were synthesized by the insertion of zinc oxide nanoparticles in the pristine graphene nanoplatelets, using a novel nanotechnological process based on patented procedures (Patent number ES2678419A1). Briefly, pristine graphene nanoplatelets were dispersed in malonic acid (molar rario 1:6), in which the starting ZnO had been previously dissolved. After homogenization, ZnO nanoparticles were precipitated between graphene nanoplatelets with a basic solution (NaOH 2.5 M) under controlled parameters, such as vigorous agitation, temperature (50 °C) and nanomaterial proportion according to the weight ratio. Bare graphene, as well as pure ZnO, were also prepared to have reference samples.

The different nanomaterials were deposited on commercial alumina (Al_2_O_3_) substrates via the spray coating method while heating the substrate to 70 °C. Figure 1 depicts a detailed representation of this substrate, which consist of two sides with an overall length and width of 25.4 mm × 4.2 mm, respectively. The top side contains platinum screen-printed interdigitated electrodes (7 mm × 3.5 mm), where the sensitive layers were deposited, whereas the bottom side includes a platinum heater for increasing the operating temperature. Nevertheless, in this paper, this heating element was not used, since the objective was to analyze sensor performance at room temperature. After that, these prepared sensors were tested for the detection of different toxic gases such as NO_2_, CO, H_2_, H_2_S, ethanol and NH_3_.

### 2.2. Material Characterization and Gas Sensing Measurements

The surface area of the hybrid nanomaterial was determined from nitrogen adsorption–desorption isotherms at 77 K with a Quadrasorb SI Model 4.0 (Quantachrome Instruments, Boynton Beach, FL, USA). Samples were outgassed at 423 K for 12 h under vacuum (6 mTorr) to eliminate chemisorbed volatiles before the adsorption isotherm was measured. Surface areas were calculated using the Brunauer-Emmett-Teller theory (BET) method.

Morphological characterization of the nanocomposite was conducted through a Transmission Electron Microscopy (TEM) (JOEL 1011, Akishima, Japan) operated at 100 kV, and the images were taken on the powder samples without any previous treatment.

The two obtained sensors were characterized using different techniques, such as Raman via a Raman spectrometer (Renishaw, plc., Wotton-under-Edge, UK), with a laser wavelength of 514 nm to check the crystallinity of the materials, and a Field Emission Scanning Electron Microscope (FESEM) using a Carl Zeiss AG-Ultra 55 (ZEISS, Jena, Germany) to study the surface morphology and to check the distribution of the NPs on the graphene layer and Photoluminiscece (PL) measurements to analyze the defects in the sensing layer. These later measurements were performed at room temperature using a chopped Kimmon IK Series He-Cd laser (325 nm and 40 mW). Fluorescence was dispersed with an Oriel Corner Stone 1/8 74,000 monochromator, detected using a Hamamatsu H8259-02 with a socket assembly E717-500 photomultiplier, and amplified through a Stanford Research Systems SR830 DSP. A filter in 360 nm was used to stray light. All spectra were corrected for the response function of the setups. FESEM was also used to determine the deposited nanocomposite layer thickness using a cross-sectional view.

Gas sensing measurements were conducted by placing the different sensors in an airtight Teflon chamber with a volume of 35 cm^3^. A continuous stream of dry air (Air Premier, 99.995% purity) and diluted gases were passed through the testing chamber with a total flow of 100 mL/min via a set of Bronkhorst mass-flow controllers. The target gases from calibrated bottles (NO_2_-1 ppm, CO-100 ppm, NH_3_-100 ppm, H_2_S-100 ppm, H_2_-1000 ppm and ethanol-20 ppm balanced in dry air) were further diluted by means of the mass flow controllers set. The resistance changes were continuously acquired using an Agilent HP 34972A multimeter. The humidity effect on the sensing performance was assessed by humidifying the gas stream through a controller evaporator mixer (CEM). The sensing responses were calculated using this formula R (%) = ((R_g_ − R_a_)/R_a_) × 100, where R_g_ and R_a_ correspond to the resistance level after and before gas exposure, respectively.

## 3. Results

### 3.1. Material Characterization

Figure 2a shows the Raman spectra of the G95/5 sensor. Only graphene specific peaks were noticed at shifts of 1347 cm^−1^ for the D-band, 1576 cm^−1^ for the G-Band, and the 2D band at 2696 cm^−1^. Despite the presence of the ZnO in the graphene, the peak corresponding to the NPs is absent in the spectrum due to its low concentration of 5%. The same graphene peaks were observed also in Figure 1b for the G80/20 sensor with the D-band, G-band, and 2D band shifts at, respectively, 1351 cm^−1^, 1579 cm^−1^, and 2704 cm^−1^, although in this spectrum a new peak is present at 415 cm^−1^, which can be attributed to the ZnO nanoparticles since its specific peak usually appears at around 430 cm^−1^. These typical peak positions were widely reported for graphene samples, but peak intensities vary depending on the crystallinity of the samples [29]. Specifically, D-band is related to disorders and defects in the graphene lattice and can be also used to determine the shape of the edge of the graphene flake [30]. Both Figure 2a,b show a slightly intense D-band, which means the disorder level is a bit high, and the edge of the graphene flake is armchair shaped. Regarding the G-band, this peak represents the planar configuration sp^2^ bonded carbon, and by performing a polarization study of the band under uniaxial strain, it is possible to determine the orientation of the graphene on the substrate [31]. Additionally, the G-band position reveals information about the thickness of the layer [32]; specifically, thicker layers tend to shift the G-band to lower energies. Thereby, Figure 2a,b show the peak at around 1580 cm^−1^, revealing that a thin graphene layer was achieved [33]. In addition, the intensity of the G- and D-band peaks confirms the good crystallinity of both studied graphene materials. Therefore, in both Figures, an intense 2D band is shown, but still less intense than the disorder bands; the graphene obtained is multi-layered [32].

The obtained FESEM images of the layers present on the surface of the G95/5 and G80/20 sensors using a back-scattered electron detector (BSE) shows a black layer which is the graphene layer, big grey chunks corresponding to the alumina substrate surface, and the zinc oxide nanoparticles can be detected by the bright spots. Some ZnO nanoparticles can be seen in Figure 3a since the ZnO concentration in the material (G95/5) is only 5 wt.%. Meanwhile, in Figure 3b, the whole surface of the layer of the G80/20 sensor is homogenously covered with ZnO nanoparticles and graphene.

EDX analysis with a quantitative elemental analysis can be found in the Appendix A. The cross-sectional view of the deposited layers shows that their thickness is not homogeneous (i.e., it ranges from 1.6 to 3.9 µm) due to surface roughness of the alumina substrate.

For the G95/5 sample, a surface area of 392.734 m^2^/g was obtained by means of BET analysis. In Figure 4, one can observe the TEM images obtained for this nanocomposite. A spherically shaped dark black nanoparticle on a sheet of graphene can be clearly seen in Figure 4a, which corresponds to a ZnO nanoparticle at a magnification of 500 K. Figure 4b also exhibits a typical morphology similar to graphene-based materials with a clear vision of the numerous ZnO NPs (dark black spherical nanoparticles) decorated on the graphene sheets at a magnification of 80 K.

The quantity and type of defects in ZnO can be estimated based on PL measurements [33]. Figure 5 shows the PL results for the ZnO-loaded graphene G95/5 sample. By pumping at 325 nm, two emission bands were observed at room temperature. A near band edge (NBE) emission in the UV at around 390 nm is associated with exciton recombination processes [34], and a broader deep level (DL) defect emission band in the visible range from 480 to 630 nm [35]. The DL broad emission band showed the maximum emission intensity at around 520 nm. Defects responsible for this peak at around 520 nm are related to oxygen vacancies [35].

### 3.2. Gas Sensing Results

Since the hourly limit of exposure to NO_2_ gas according to the European Environment Agency is 106 ppb (200 ug/m^3^), gas measurement tests of different dilutions of this target gas were conducted in the range of 50 to 500 ppb at room temperature. The ZnO sensor operated at room temperature exhibited a high resistance (in the range of 10 MΩ), leading to very noisy behavior. In this case, no noticeable response was obtained in the concentration range studied. Figure 6a reflects the responses to nitrogen dioxide at the considered concentration range for the different sensors working at room temperature. It can be seen that both bare graphene and the G80/20 sample do not provide a significant response for concentrations below 500 ppb. For this concentration, bare graphene exhibits a higher response than that of the G80/20 sample. Therefore, the inclusion of this high concentration of ZnO did not lead to a better performance. In contrast, sample G95/5 outperforms the other sensors for all the concentration range tested. Then, the inclusion of a lower amount of ZnO is a good option, allowing to reach a relatively high sensitivity of 2.6% ppm^−1^ in this concentration range. Figure 6b shows the electrical resistance changes when the sensor is exposed to three different concentrations of NO_2_ in dry conditions. As expected, the figure shows a p-type behavior of the sensor, with the resistance decreasing when put in contact with NO_2_, which is an oxidizing gas. As can be seen in Figure 6b, sensor responses were rather slow, as usual when resistive sensors are operated at room temperature. Consequently, the sensor does not recover the baseline resistance even after being exposed to clean air for more than four minutes. Nevertheless, this issue could be solved using a temperature pulse or a UV light pulse, as proposed in the literature, in order to accelerate the desorption of NO_2_ from the gas sensitive surface, thus fully regaining its baseline.

Since the G95/5 sensor was able to efficiently detect very low concentrations of NO_2_, it was the subject of further measurements closer to real conditions (in other words under humidity), to check the effect of ambient moisture on sensor performance. Two different relative humidity percentages of 20 and 70% were applied, while doing the gas tests towards different concentrations of NO_2_ of 50, 150, 250, 350 and 500 ppb at room temperature. As can be seen in Figure 7, the calibration curves show a quite linear behavior in the concentration range considered, allowing us to determine a constant sensitivity as the slope of the regression lines. The values obtained are summarized in Table 1.

One can see that increasing the moisture level, the sensitivity of the senor is increased.

The limit of detection (LOD) was calculated for the three cases using the equation:LOD = 3Sy/b 

Here, Sy represents the standard deviation of y-residuals obtained from the calibration curves, while b corresponds to the slope. The sensor exhibited a LOD of 21.9 ppb for NO_2_ under dry conditions. However, the LOD increased to 47 ppb when the relative humidity was increased to 20% and to 36.9 for 70% humidity.

These results imply that in a practical situation this sensor should be operated jointly with a humidity sensor, in order to enable using the correct calibration curve for determining NO_2_ concentration.

To have an idea of the position of this work compared to the literature, the sensing performance of the G95/5 sensor was compared with previously reported sensors and gathered in Table 2. This work with a response towards 0.5 ppm of NO_2_ of 5.1% seems to be better than the works presented in the references [36,37] with responses of 3.6% and 1.4%, respectively, towards higher concentrations of the target gas, 100 times higher in the case of the work referenced [36]. Meanwhile, the other works referenced [38,39] have presented higher response than this work, since the concentration of the NO_2_ tested are way higher: 10 times higher for [38] and 200 times higher for [39].

The selectivity of the sensors was evaluated by repeating the same experimental conditions when detecting NO_2_ for detecting other target gases. In this regard, 500 ppb of NO_2_, 200 ppm of H_2_, 20 ppm of H_2_S and ethanol 25 ppm of CO and 50 ppm of NH_3_ were detected at room temperature. As can be seen in Figure 8, the inclusion of the ZnO nanoparticles results in an increased response towards nitrogen dioxide gas with a response of 0.9% and a concentration of 500 ppb. Meanwhile, the response towards carbon monoxide, hydrogen sulphide, ethanol and ammonia is very low (lower than 0.2) compared to NO_2_, although the concentration of those gases is more than 40 times higher. The highest response is obtained for hydrogen, but is less than half the one obtained for NO_2_, due to the concentration being 200 times higher. This proves the high selectivity of this sensor to nitrogen dioxide gas (NO_2_).

Finally, to check the long-term stability of the sensors, measurements shown in Figure 6 were repeated nine months late. The comparative results are shown in Figure 9. As can be seen the sensors show a quite suitable long-term stability.

### 3.3. Gas Sensing Mechanism

Bare graphene is a mild p-type nanomaterial, and when gas molecules get adsorbed on its large surface area, the local carrier concentration quickly changes, inducing resistance changes. In the case of NO_2_, which is a strong oxidant gas (electron-withdrawing gas), the interaction between the graphene layer and the gas results in a decrease in the sensor resistance. Conversely, when detecting reducing gases, such as NH_3_ (electron donor gas), the sensor resistance increases [40].

Considering the use of p-type graphene loaded with n-type ZnO nanoparticles, it is expected that a p-n junction will form on the contact surface. Specifically, electrons from the ZnO conduction band probably flows to the graphene, resulting in a depletion layer at the interface of the two materials. As a result, ZnO@Graphene films with a ratio of 20/80 exhibit a higher resistance baseline than their 5/95 counterparts. This is because the ZnO nanoparticles transfer electrons to the graphene, reducing the concentration of majority carriers in p-type graphene. Consequently, the conductivity of the sensing film is reduced.

When ZnO@Graphene is exposed to air, oxygen molecules capture electrons from the valence band and become adsorbed onto the sensor surface (1).
O_2_ (g) + 2e^−^ → 2O_(ads.)_^−^(1)

When NO_2_ molecules interact with the sensor surface, two distinct reactions may occur. Firstly, NO_2_ can adsorb onto the hybrid nanocomposite (2). Secondly, NO_2_ may interact with adsorbed oxygen species, which can lead to changes in the resistivity of the sensitive film (3). In both cases, NO_2_ adsorption captures free electrons from the sensitive film, leading to a significant increase in the conductivity [41].

However, the p-n junction also enhances the sensor’s sensitivity by expanding the depletion region when the film interacts with NO_2_ molecules [42]. Figure 10 illustrates the potential band diagram for the gas sensing mechanism.
NO_2_ (g) + e^−^ → NO_2(ads.)_^−^(2)
(3)$$2{\mathrm{NO}}_{2}(\text{ads.})+O_{2^{-}}+e^{-}\to 2{\mathrm{NO}}_{3}^{-}$$

It is interesting to note that G95/5 demonstrates a higher sensing performance than G80/20. In other words, a higher content of ZnO does not necessarily translate into better sensitivity, despite the higher flow of electrons from ZnO to graphene. This is probably because a lower amount of nanoparticles brings graphene sheets closer and better interconnected. As a result, the charge transfer is more efficient compared to higher amounts of nanoparticles, which may obstruct the conduction paths along the graphene sheets [43].

Finally, it is worth noting that ambient moisture usually has an enhancing effect on the sensitivity of graphene-based sensors [44]. Considering the room temperature detection, the water molecules probably act as a mediated adsorption site for NO_2_, causing an increase in sensitivity towards the target gas [45].

## 4. Conclusions

This work reports the fabrication of a graphene-based sensor, decorated with ZnO nanoparticles using a novel and patented nanotechnological method for enhanced NO_2_ sensing at room temperature. The synthesis method has been already scaled up for the mass production of gas sensitive nanomaterials (from few g up to kg). ZnO(5%)-loaded graphene sensors clearly outperform pure graphene or pure ZnO sensors when operated at room temperature. The loading of the graphene with two different percentages of ZnO has shown that the lower the loading, the better the sensing capabilities of the sensor, since the G80/20 was not capable of detecting NO_2_ at concentrations lower than 250 ppb. Meanwhile, G95/5 could detect as low as 50 ppb of the target gas. Furthermore, the G95/5 sensor showed very good responses, stability, and reproducibility in the range of 50-250-500 ppb of NO_2_ The sensing performance becomes significantly enhanced under a humid atmosphere to reach almost five-fold, the response under a dry atmosphere, which makes it a very promising material for NO_2_ detection for industrial use in farms or factories.

## Figures and Tables

**Figure 1 sensors-23-06055-f001:**
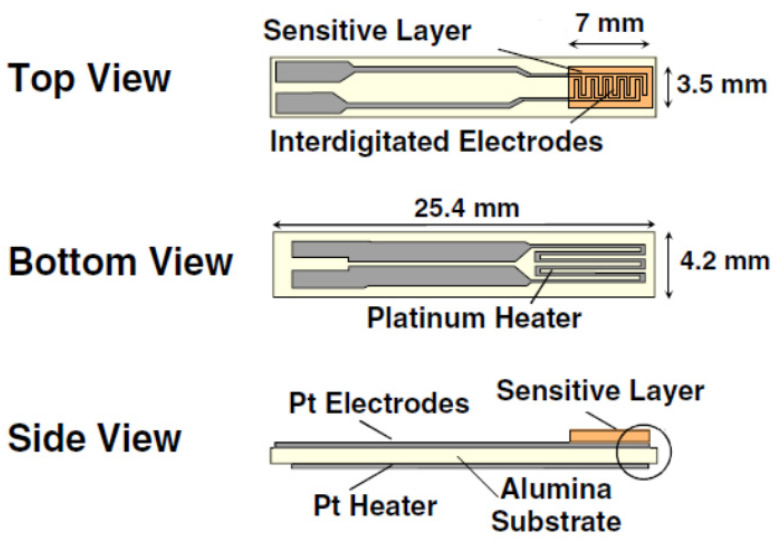
The planar structure of the top, bottom, and side view of the alumina substrate used for gas sensing measurements.

**Figure 2 sensors-23-06055-f002:**
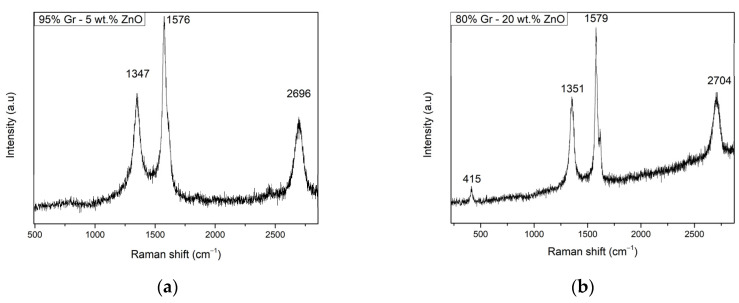
Raman spectra of (**a**) G95/5 sensor and (**b**) G80/20 sensor.

**Figure 3 sensors-23-06055-f003:**
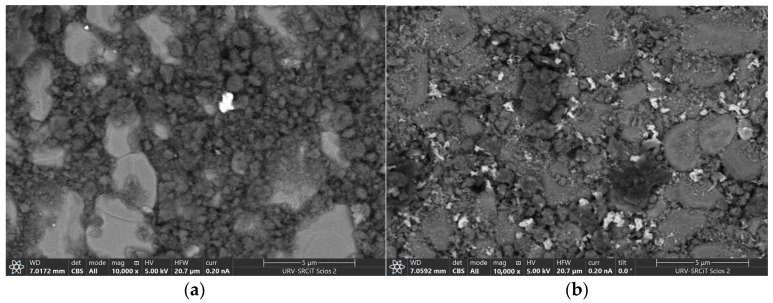
FESEM images of (**a**) G95/5 surface exhibiting few nanoparticles of ZnO (**b**) G80/20 showing a homogenous distribution of the ZnO nanoparticles on the surface of the layer.

**Figure 4 sensors-23-06055-f004:**
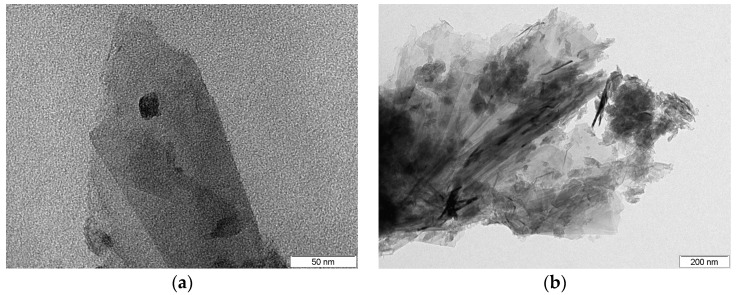
TEM images of (**a**) a single spherical shaped ZnO nanoparticle on graphene sheets and (**b**) multiple ZnO nanoparticles on graphene sheets.

**Figure 5 sensors-23-06055-f005:**
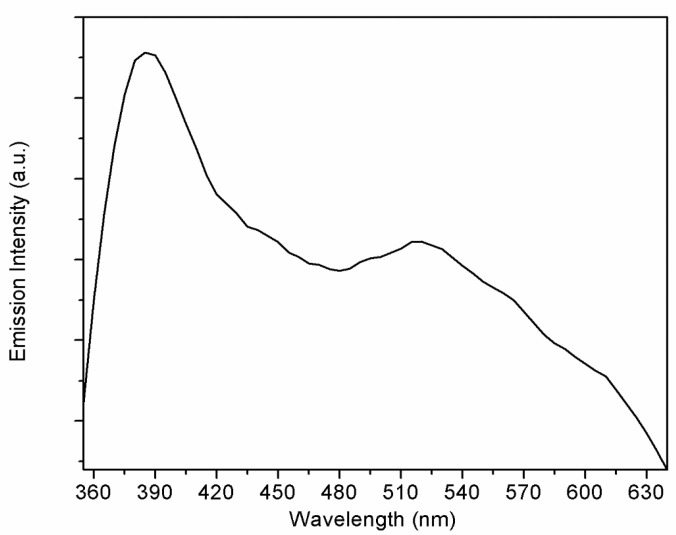
Photoluminescence spectrum for the G95/5 sample, recorded at room temperature.

**Figure 6 sensors-23-06055-f006:**
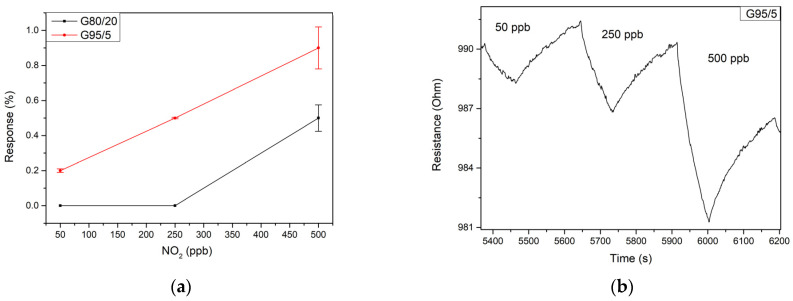
(**a**) Responses of the G95/5 (red) and G80/20 (black) sensors at room temperature for 50, 250 and 500 ppb of NO_2_ (**b**) resistance changes of the sensor G95/5 towards different dilutions of NO_2_ at room temperature.

**Figure 7 sensors-23-06055-f007:**
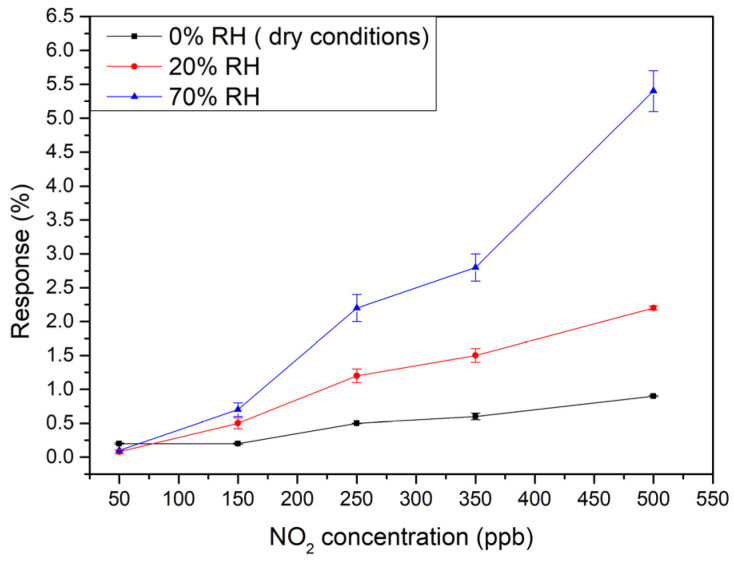
Calibration curves of G95/5 sensor under different relative humidity levels: 0 (dry), 20%, and 70%.

**Figure 8 sensors-23-06055-f008:**
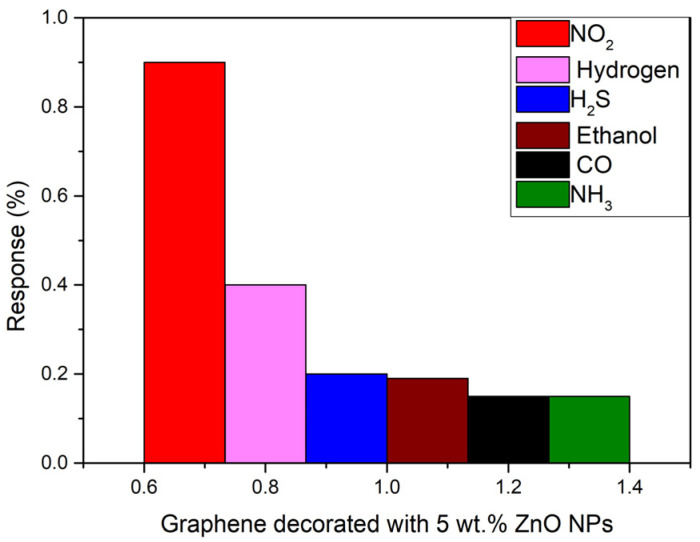
Comparison chart of the response of the sensors towards 500 ppb of NO_2_ (red), 200 ppm of H_2_ (pink), 20 ppm of H_2_S (blue), 20 ppm of ethanol (brown), 25 ppm of CO (black), and 50 ppm of NH_3_ (green) at room temperature and dry conditions.

**Figure 9 sensors-23-06055-f009:**
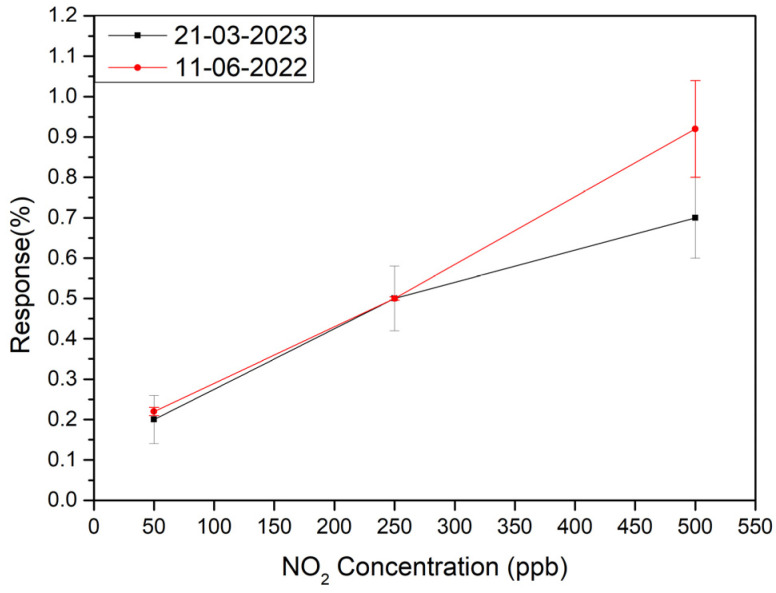
Long term stability test of the sensitive layer of 9 months apart.

**Figure 10 sensors-23-06055-f010:**
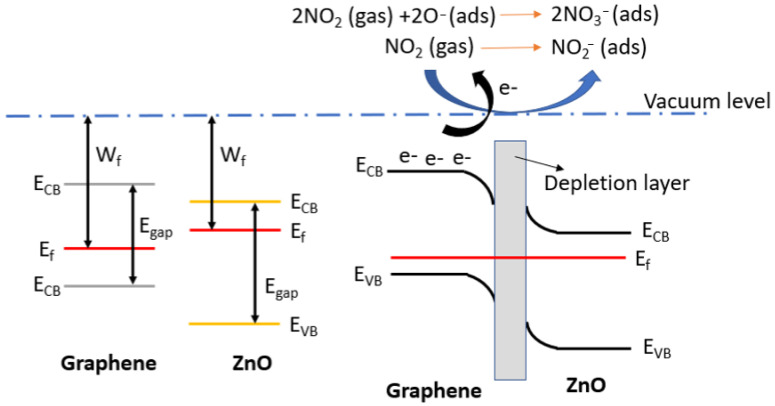
Band diagram of the p-n junction between graphene and ZnO. When the gas sensor interacts with NO_2_, this gaseous compound captures the free electrons, leading to significant resistance changes due to the decrease of minority carriers in the graphene.

**Table 1 sensors-23-06055-t001:** Sensitivity values of the G95/5 sensor under dry and different humid conditions.

Relative Humidity (%)	Sensitivity (%·ppm^−1^)
0	1.6
20	4.7
70	11.7

**Table 2 sensors-23-06055-t002:** Responses of different graphene decorated ZnO-based sensors cited in the literature compared to the present work.

Composition	Temperature (°C)	NO_2_ Concentration (ppm)	Response	Ref.
Graphene decorated with 5 wt.% ZnO (70% RH) (GNANOMAT)	RT	0.5 (500 ppb)	5.1%	This work
ZnO/graphene aerogel	RT	50	3.6%	[36]
rGO/ZnO flowers and nanoparticles	RT	1.5	1.4%	[37]
rGO/ZnO laser modified	RT	5	6.2%	[38]
rGO/ZnO nanorods	RT	100	17.4%	[39]

## Data Availability

Data used in this paper is available upon demand.

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
