# Peer review of "ZnO-Loaded Graphene for NO2 Gas Sensing"

_sensors, 2023, doi:10.3390/s23136055_

Round 1

Reviewer 1 Report

Recommendation: Major revisions are needed as noted.

In this article, M.A. Alouani et al., investigated the NO2 gas sensing properties using ZnO-loaded graphene nanocomposite at room temperature at ppb level. Although graphene/ZnO-based nanocomposites sensing material for chemosensitive sensors is not a new concept, but the manuscript is also nicely written. However, they failed to explaining the characterization techniques to explain their sensing materials mechanism. There are many serious concerns that need to be addressed before it can be processed for further steps in the publication in Sensors journal. 

1)       Novelty of your works was further enriched in revised manuscript. About the choose of 2D based-graphene/ZnO MOXs. What’s new in your material choosing. Explain elaborately and how it could be useful for NO2 gas.

2)       SEM images are not clear and visible to distinguish exact morphology. Since morphology also plays a key role in gas sensing. Also, use your own scale bars.

3)       What is the thickness of the sensing layer. If possible provide TEM analysis or provide FESEM high magnification (highly recommended) images. What is the sensing layer coated area?

4)       Have the authors tested the pristine counterparts sensing performances (pristine graphene and ZnO separately) sensing layer. Provide comparative sensing results to prove the potential of your nanocomposite.

5)       How did the authors estimate the response/recovery time?

6)       What is the detection limit of your sensor?

7)       What is the elemental percentage of each element from the EDS mapping spectra.

8)       Since it was an oxide semiconducting material. Oxygen vacancies (OV) are vital. Please provide XPS analysis. Since OV plays a key role in gas adsorption ZnO/graphene nanocomposite material. Please explain using XPS analysis.

9)       For the selectivity test 3 gases are not enough. please provide at least 5 gases and compare your gas sensing study.

10)   What is the surface are of the nanocomposite.

11)   Provide cycling and long-term stability test results.

12)   Authors should add their sensor response in Table 1 with more literature along with error analysis.

13)   Mechanism is the core part and it seems to be very poor in this study please take care of it. What is the physical and chemical significance? As per the authors discussion graphene is p-type and ZnO is n-type in that is the nanocomposite posses a p-n junction or in my opinion based the synthesis process and work function of graphene it also can act as metal in that case graphene/ZnO can act as Schottky/ohmic junction. Please enlarge your good sensing results with enriched mechanism. Explain detailly. Authors should explain the sensing mechanism using suitable band analysis.

14)   Authors should please prove a more detailed analysis of the effect of humidity on your enhanced sensor gas sensing properties.

15)   Some of the references used in the manuscript are not up-to-date; authors should cite the following references:  10.1016/j.snb.2022.133140, 10.1039/D2NJ04117K, 10.1016/j.jhazmat.2021.128174,

Author Response

We want to thank the reviewer for his/her helpful comments. Below are the answers to the questions that have been formulated.

In this article, M.A. Alouani et al., investigated the NO2 gas sensing properties using ZnO-loaded graphene nanocomposite at room temperature at ppb level. Although graphene/ZnO-based nanocomposites sensing material for chemosensitive sensors is not a new concept, but the manuscript is also nicely written. However, they failed to explaining the characterization techniques to explain their sensing materials mechanism. There are many serious concerns that need to be addressed before it can be processed for further steps in the publication in Sensors journal. 

  • Novelty of your works was further enriched in revised manuscript. About the choose of 2D based-graphene/ZnO MOXs. What’s new in your material choosing. Explain elaborately and how it could be useful for NO2 gas.

We have included a paragraph in the introduction to point out the novelty of the work.

  • SEM images are not clear and visible to distinguish exact morphology. Since morphology also plays a key role in gas sensing. Also, use your own scale bars.

New FESEM images as well as TEM images of the powder have been obtained and included in the new version of the paper. Also, EDS mappings have been included as supporting information to help interpreting the images.

3)       What is the thickness of the sensing layer. If possible provide TEM analysis or provide FESEM high magnification (highly recommended) images. What is the sensing layer coated area?

We have substituted the FESEM images with new ones with higher resolution and we have also included TEM images of the powder. We have used FESEM cross-sectional view to determine the layer thickness, which is not homogeneous, ranging from 1.6 to 3.9 µm

Regarding the coated area, it is shown in Figure 1. The area is 7 mm x 3.5 mm.

4)        Have the authors tested the pristine counterparts sensing performances (pristine graphene and ZnO separately) sensing layer. Provide comparative sensing results to prove the potential of your nanocomposite.

Since the test were performed at room temperature, we did not obtain any response for pristine ZnO. On the other hand, with the 100% graphene sensor we could not obtain any relevant response for concentrations lower than 500 ppb. For this concentration the response was 0.7.

5)       How did the authors estimate the response/recovery time?

As usual when working at low temperatures, the response/recovery times are quite large. Therefore, we have not waited the sensor to reach a steady state regime to allow a reliable response time determination. That’s why we have not included this information in the paper. Nevertheless, we have commented the large response/recovery times showed by these sensors.

6)       What is the detection limit of your sensor?

The limit of detection (LOD) was calculated for both dry and humid atmospheres. The LOD was determined using the equation:

LOD = 3Sy/b

Here, Sy represents the standard deviation of y-residuals obtained from the calibration curves, while b corresponds to the slope. The sensor exhibited a LOD of 21.9 ppb for NO2 under dry conditions. However, the LOD increased to 47 ppb when the relative humidity was increased to 20% and to 36.9 for 70% humidity

7)       What is the elemental percentage of each element from the EDS mapping spectra.

The EDS mapping not only includes the elements of the nanocomposite (carbon, zinc and oxygen), but also other elements coming from the alumina substrate, as well as traces of other elements that can be considered contaminants or due to a bad assignment of the system (Si and Mg). Regarding the main elements that can be related to the sensing nanocomposite (carbon and zinc), the normalized weight percentages are 8.44 and 12.68 respectively for the G80/20 sample and 37.70 and 2.17 for the G95/5 sample. We have included this information in the supporting information.

8)       Since it was an oxide semiconducting material. Oxygen vacancies (OV) are vital. Please provide XPS analysis. Since OV plays a key role in gas adsorption ZnO/graphene nanocomposite material. Please explain using XPS analysis.

Photoluminescence (PL) measurements can also show the type of defects in ZnO, and it was observed a peak at around 520 nm related to oxygen vacancies. We have included this analysis in the text.

9)       For the selectivity test 3 gases are not enough. please provide at least 5 gases and compare your gas sensing study.

We have included 5 interfering gases in the selectivity analysis.

10)   What is the surface are of the nanocomposite.

The surface area of the G95/5 sample, obtained from BET measurement, is 392.734 m²/g. This information has been included in the paper.

11)   Provide cycling and long-term stability test results.

We repeated the NO2 measurements in dry air after 9 months and the results were very similar. We have included this information in the paper.

12)   Authors should add their sensor response in Table 1 with more literature along with error analysis.

We decided to show the results with humidity in another way that we think points out how we have determined the sensitivity values. Since we have repeated the experiments again to show at least 5 calibration points as requested, the sensitivity values in the new version are slightly different from the original submission.

13)   Mechanism is the core part and it seems to be very poor in this study please take care of it. What is the physical and chemical significance? As per the authors discussion graphene is p-type and ZnO is n-type in that is the nanocomposite posses a p-n junction or in my opinion based the synthesis process and work function of graphene it also can act as metal in that case graphene/ZnO can act as Schottky/ohmic junction. Please enlarge your good sensing results with enriched mechanism. Explain detailly. Authors should explain the sensing mechanism using suitable band analysis.

We have changed completely the discussion of the working mechanism, including a band analysis image as requested.

14)   Authors should please prove a more detailed analysis of the effect of humidity on your enhanced sensor gas sensing properties.

The baseline resistance level shows a gradual increase with an increase in relative humidity percentage. This can be attributed to water molecules acting as electron donors. However, additional research is necessary to gain a comprehensive understanding of the ongoing reaction processes.

15)   Some of the references used in the manuscript are not up-to-date; authors should cite the following references:

https://doi.org/10.1016/j.snb.2022.133140

https://doi.org/10.1039/D2NJ04117K

https://doi.org/10.1016/j.jhazmat.2021.128174

We have included the requested references in the introduction.

Reviewer 2 Report

In this work, the authors reported a ZnO loaded graphene for NO2 gas sensing. However, there are many problems in the motivations, experimental, and data results. Besides, the gas sensing performances are incomplete. The manuscript may be accepted after major modifications.

1.    Introduction: 1) Please emphasize the importance of NO2 detection in the first paragraph. 2) The selection strategy of ZnO as sensing materials should be explained in details. The band gap of 3.37 eV and exciton binding energy of 60 meV are not the key reasons in choosing sensing materials. 3) Now that the authors admit the existence of reported gas sensors based on ZnO and graphene, what’s the innovation of this work? 4) “…graphene was also decorated with ZnO nanoparticles using a novel nanotechnological process to…” What’s the novel technology? 5) Recently, many room-temperature NO2 sensors based on metal oxides and their composites have been reported. Therefore, recent advances of them should be emphasized before introducing the reasons for selecting ZnO as sensing material. Such as: J. Hazard. Mater. 434 (2022) 128836; Sensors 21 (2021) 8269.

2.    Experimental: 1) What’s the organic acid in this work? Experiment’s details should be given. For example, the concentration of organic acid, the amount of graphene, dissolved ZnO, and basic solution, and the details of the controlled parameters (such as agitation, temperature, or nanomaterial proportion). 2) The synthesis process is confusing, why not directly disperse the graphene nanoplatelets into the zinc ions solution? 3) What’s the concentration of the original gas sources (NO2, NH3, and CO)?

3.    How to control the crystallization process and further obtain the ZnO nanoparticles with different grain sizes? More details and comparison results should be given to optimize the synthesis process.

4.    Gas sensing performances: 1) More than 5 different concentrations should be measured to obtain a calibration curve. 2) Many parameters, such as repeatability, response/recovery times, and long-term stability are missing in this work. 3) Please provide the real-time data of the gas sensing performances under different RH values. Besides, what’s the effect of humidity on the base resistances of the sensor? 4) Two gases (NH3 and CO) are not enough to evaluate the selectivity of the sensor.

5.    Mechanism: “…When exposed to oxidizing gases such as NO2, the gas molecules adsorb directly onto ZnO and react with the oxygen ions, leading to the formation of NO3-, which will eventually trap electrons from the graphene surface.” The explanations are confusing. Detail chemical reaction formula should be supplemented. Furthermore, if the NO2 can react with the oxygen ions, what’s the relationship between the further released electrons (from the oxygen ions) onto the ZnO and the trapped electrons from the graphene surface?

6.    English writing is not standardized, for example, the numbers in the chemical formula need subscripts (including references).

7.    List of references: Most of the references are out of date (before 2014). Room temperature NO2 gas sensors and sensing materials (such as MXene, metal oxide composites, Telluride) have developed rapidly. It is recommended to cite the literature of the last three years.

8.    Check the format of references.

Author Response

We want to thank the reviewer for his/her helpful comments. Below are the answers to the questions that have been formulated.

Comments and Suggestions for Authors

In this work, the authors reported a ZnO loaded graphene for NO2 gas sensing. However, there are many problems in the motivations, experimental, and data results. Besides, the gas sensing performances are incomplete. The manuscript may be accepted after major modifications.

  1. Introduction:

1.1) Please emphasize the importance of NO2 detection in the first paragraph.

We have emphasized the relevance of NO2 detection in the introduction including new references.

1.2) The selection strategy of ZnO as sensing materials should be explained in details. The band gap of 3.37 eV and exciton binding energy of 60 meV are not the key reasons in choosing sensing materials.

Certainly, those characteristic of ZnO were not the key reason for choosing ZnO. The reasons are listed in the introduction later: good response to different toxic gases (including NO2), chemical stability, which is relevant to assure good long term stability of the sensors, and it is cheap and environmental friendly. Moreover, some previous reports were available to compare our results.

1.3) Now that the authors admit the existence of reported gas sensors based on ZnO and graphene, what’s the innovation of this work?

The innovation of the work relies on the synthesis procedure. A new synthesis, protected by a Spanish patent, was used to obtain the nanocomposite. This procedure uses low temperatures and pressures to reach the desired nanocomposite and can be used to obtain big amounts of material, being an industrial process. To assess the performance of this approach we have compared our results with the ones reported in the literature for similar materials.

1.4) “…graphene was also decorated with ZnO nanoparticles using a novel nanotechnological process to…” What’s the novel technology?

As explained in the previous question, we have used a synthesis route using low temperatures/pressures, producing the material in high amounts achieving reasonably good results.

1.5) Recently, many room-temperature NO2 sensors based on metal oxides and their composites have been reported. Therefore, recent advances of them should be emphasized before introducing the reasons for selecting ZnO as sensing material. Such as: J. Hazard. Mater. 434 (2022) 128836; Sensors 21 (2021) 8269.

New references have been included in the introduction following this suggestion.

  1. Experimental:

2.1) What’s the organic acid in this work? Experiment’s details should be given. For example, the concentration of organic acid, the amount of graphene, dissolved ZnO, and basic solution, and the details of the controlled parameters (such as agitation, temperature, or nanomaterial proportion).

The organic acid employed was malonic acid in water at a molar ratio of 1:6. The basic solution was 2.5M NaOH, and the synthesis was done under vigorous agitation and temperature (50 ºC).

The percentages of graphene/ZnO in the material are by weight so, this is the amount of graphene and ZnO nanoparticles.

2.2) The synthesis process is confusing, why not directly disperse the graphene nanoplatelets into the zinc ions solution?

Firstly, the zinc oxide was dissolved in malonic acid and then the graphene was homogeneously dispersed in the Zn solution to promote the precipitation of ZnO nanoparticles on it. In this way, the aim was to reduce the agglomeration of the ZnO nanoparticles, avoid the re-stacking of the graphene sheets, obtain a larger active surface area and a good contact between ZnO and graphene.

2.3) What’s the concentration of the original gas sources (NO2, NH3, and CO)?

The original concentrations for these gases were 1 ppm for NO2 and 100 ppm for the others. We have included this information in the paper, as well as, the concentrations of the bottles for the new gases used for the selectivity analysis

  1. How to control the crystallization process and further obtain the ZnO nanoparticles with different grain sizes? More details and comparison results should be given to optimize the synthesis process.

The crystallization process can be controlled by the rate of addition of the NaOH, the temperature in the precipitation, the stirring speed or the digestion time. All parameters are key in the crystallization process, but also in the scale up process.

  1. Gas sensing performances:

4.1) More than 5 different concentrations should be measured to obtain a calibration curve.

We have used 5 measurements at 5 different concentrations to show the calibrations curves both in dry conditions and at two different levels of humidity (Figure 7)

4.2) Many parameters, such as repeatability, response/recovery times, and long-term stability are missing in this work.

We have included the long-term stability test results (Figure 9) comparing the results of the G95/5 sensor after more than 9 months. We have also included a comment on the repeatability of the procedure comparing the results of two different sensors based on the same powder: G95/5. Nevertheless, since the sensors are quite slow in their response and recovery, we did not wait for the sensor reaching a steady state, what affects the calculation of the response and recovery times. Therefore, we decided not including this information and just include a comment about their slow response, which can be seen in Figure 6b.

4.3) Please provide the real-time data of the gas sensing performances under different RH values. Besides, what’s the effect of humidity on the base resistances of the sensor?

The baseline resistance level shows a gradual decrease with an increase in relative humidity percentage. However, additional research is necessary to gain a comprehensive understanding of the ongoing reaction processes.

4.4) Two gases (NH3 and CO) are not enough to evaluate the selectivity of the sensor.

We have included up to 5 interfering gases for the selectivity test

  1. Mechanism: “…When exposed to oxidizing gases such as NO2, the gas molecules adsorb directly onto ZnO and react with the oxygen ions, leading to the formation of NO3-, which will eventually trap electrons from the graphene surface.” The explanations are confusing. Detail chemical reaction formula should be supplemented. Furthermore, if the NO2 can react with the oxygen ions, what’s the relationship between the further released electrons (from the oxygen ions) onto the ZnO and the trapped electrons from the graphene surface?

We have changed completely the explanation of the working mechanism in order to clarify it, including a band diagram, as requested by another reviewer

  1. English writing is not standardized, for example, the numbers in the chemical formula need subscripts (including references).

We have revised the chemical formula along the paper

  1. List of references: Most of the references are out of date (before 2014). Room temperature NO2 gas sensors and sensing materials (such as MXene, metal oxide composites, Telluride) have developed rapidly. It is recommended to cite the literature of the last three years.

We have included more recent references.

  1. Check the format of references.

We have revised the format of the references.

Round 2

Reviewer 1 Report

The authors have answered all the queries raised by the reviewers and the manuscript was revised well. Thus, this manuscript can be given acceptance for publication in Sensors journal.

Author Response

We want to thank again Reviewer 1 for her/his valuable comments.

Reviewer 2 Report

There are still some minor issues in the revised manuscript:

1. The image quality needs to be improved. For example, line width, font size, and image size are unified (Figure 6).

2. Check the format of the references. For example. The information should be complete, the format should be unified, the journal name should be abbreviated, and the numbers in the chemical formula should be subscript. In addition, many references are outdated and it is recommended to cite the latest three years.

Author Response

Thanks again. Certainly the format of some figures was not good enough. We have homogenized the format of all the figures in the paper. It is also true that there were some problems with the format of the references. We have solved this problem.

Regarding the dates of the references, in the present form of the paper we have cited 13 references in the period 2021-23 and 10 references more if we add two years more (2019-20). These are more than half the total references of the paper. We think the former references are useful for the paper discussion and we don't want to eliminate anyone.